# Species-Specific Responses of Bird Song Output in the Presence of Drones

Andrew M. Wilson *[ID], Kenneth S. Boyle, Jennifer L. Gilmore, Cody J. Kiefer and Matthew F. Walker

Environmental Studies Department, Gettysburg College, 300 N. Washington Street, Gettysburg, PA 17325, USA; boylke02@alumni.gettysburg.edu (K.S.B.); gilmje03@alumni.gettysburg.edu (J.L.G.); kiefco01@alumni.gettysburg.edu (C.J.K.); walkma02@alumni.gettysburg.edu (M.F.W.)

\* Correspondence: awilson@gettysburg.edu

**Abstract:** Drones are now widely used to study wildlife, but their application in the study of bioacoustics is limited. Drones can be used to collect data on bird vocalizations, but an ongoing concern is that noise from drones could change bird vocalization behavior. To test for behavioral impact, we conducted an experiment using 30 sound localization arrays to track the song output of 7 songbird species before, during, and after a 3 min flight of a small quadcopter drone hovering 48 m above ground level. We analyzed 8303 song bouts, of which 2285, from 184 individual birds were within 50 m of the array centers. We used linear mixed effect models to assess whether patterns in bird song output could be attributed to the drone's presence. We found no evidence of any effect of the drone on five species: American Robin *Turdus migratorius*, Common Yellowthroat *Geothlypis trichas*, Field Sparrow *Spizella pusilla*, Song Sparrow *Melospiza melodia*, and Indigo Bunting *Passerina cyanea*. However, we found a substantial decrease in Yellow Warbler *Setophaga petechia* song detections during the 3 min drone hover; there was an 81% drop in detections in the third minute (Wald test, $p < 0.001$) compared with before the drone's introduction. By contrast, the number of singing Northern Cardinal *Cardinalis cardinalis* increased when the drone was overhead and remained almost five-fold higher for 4 min after the drone departed ($p < 0.001$). Further, we found an increase in cardinal contact/alarm calls when the drone was overhead, with the elevated calling rate lasting for 2 min after the drone departed ($p < 0.001$). Our study suggests that the responses of songbirds to drones may be species-specific, an important consideration when proposing the use of drones in avian studies. We note that recent advances in drone technology have resulted in much quieter drones, which makes us hopeful that the impact that we detected could be greatly reduced.

**Keywords:** bioacoustics; drone; noise pollution; songbird; UAV; UAS

## 1. Introduction

Drones, or Unoccupied Aircraft Systems (UASs), are now well-established tools in field ecology, especially for the purposes of mapping habitats or counting and tracking megafauna [1–4]. Drones have proven to be effective tools for conducting aerial surveys of large birds, especially those found in open habitats [5,6] or that nest or roost in tree canopies [7,8]. However, there have been few studies that show drones to be useful for monitoring small birds, or those that inhabit dense vegetation, where the reach of aerial imagery is limited. Despite some of the potential advantages of drones, such as their efficiency, ease of access, and potential to reduce disturbance [9], drones are not widely used for monitoring smaller birds, such as songbirds, which constitute the majority of bird species and are abundant in most terrestrial environments [10–12]. Recent advances coupling drones with audio-recorders for bioacoustic monitoring [13,14] and infrared cameras to detect passerine nests [15], suggest that the use of drones for monitoring smaller birds could grow.

An important drawback of drones is that they emit noise, which, in addition to their visual impact, can disturb wildlife. There have now been many studies that investigate the

impacts of drones on wildlife. A review of these concluded that smaller drones, flown at greater heights, and spending reduced time overhead, were less likely to elicit a response in studied organisms [16]. However, there are still many unknown impacts of drones, especially on non-target organisms.

In developing protocols to use drones for bioacoustics monitoring, we are interested in the possible impacts of drones on songbirds for two important reasons: firstly, if drones do impact songbird behavior, then this introduces an ethical question as to whether they should be used at all; and secondly, if drones cause songbirds to move locations or change their vocalization behavior, the data derived from airborne bioacoustics could be biased. In this study, we aim to assess the impacts of hovering a drone over a study area for the purpose of airborne bioacoustics monitoring; specifically, we test the hypothesis that hovering drones reduce bird song output. While the bounds of our study were narrow—we focused on the impact of a single drone model, flown at a specific height—we aimed to shed light on the potential impact of drones on songbirds, which to most ecologists using drones would be considered non-target species. Further, there is a wider concern that as an additional source of anthropogenic noise pollution, drones flown by recreationists as well as professionals may add to growing noise pollution problems. Our study informs this broader context by assessing the ecological impact of noise pollution from drones on bird singing behavior.

## 2. Materials and Methods

To test whether there was a change in songbird vocalizing behavior that could be attributed to the drone's presence, we set up an experiment whereby ambient bird sound in the area under the drone was recorded before, during, and after a drone flight. Traditionally, bird population surveys are completed using the point count technique, in which ornithologists record visual and auditory bird cues for a fixed period at a series of pre-determined locations [17]. Following a previous study [13], we simulated an airborne bird point count by hovering a small quadcopter (DJI Mavic Pro) over a pre-determined count location for 3 min. The traditional point count duration is typically from 3 to 10 min; we chose the lower end of this range because it minimizes flight time, thereby allowing for greater survey efficiency—potentially allowing multiple points to be surveyed on each drone battery. Additionally, shorter flight duration is less likely to cause excessive disturbance [16].

### 2.1. Field Study

We used sound localization to locate singing birds in the vicinity of a ground-based Automated Recording Unit (ARU) array to track bird song output before, during, and after our drone flights (Figure 1a). By arranging four ARUs in a 50 × 50 m array, we were able to measure the differences in the time taken for each bird vocalization to reach each recorder, a method called "time difference of arrival" (TDoA). The differences in the time of arrival allowed us to triangulate the birds in two-dimensional space inside and beyond the array, using the Sound Finder Microsoft Excel application [18]. The Sound Finder method has been proven to allow accurate localizations, with a mean error of 4.3 m, and 74% of localizations, with an accuracy of less than 10 m [18].

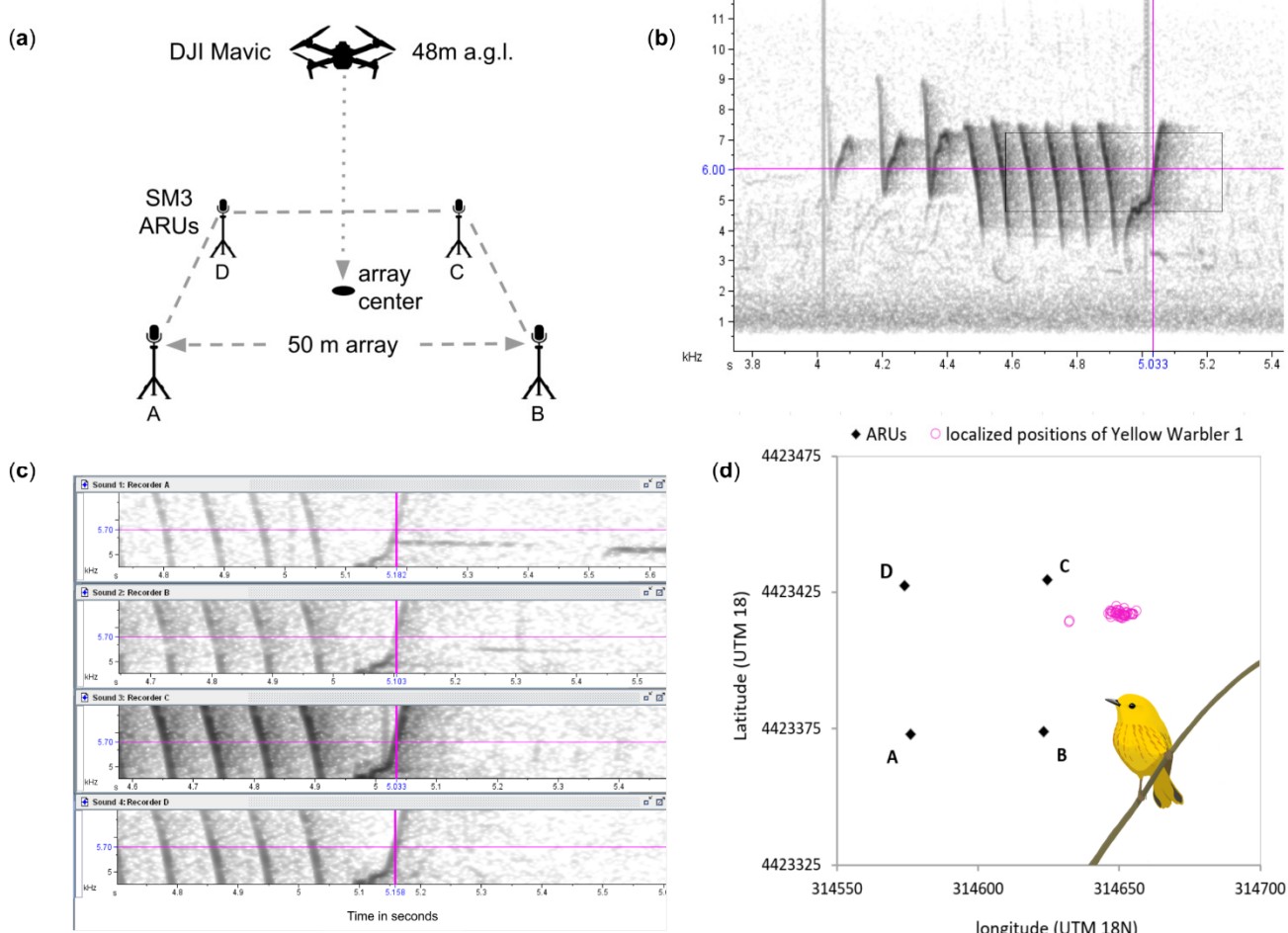

**Figure 1.** Methods. Pane (**a**) shows the setup of each experimental array, (**b**) shows the spectrogram of a Yellow Warbler song from recorder C with a black rectangle showing the inset in pane (**c**), which shows the measurement of time difference of arrival in Program Raven, and (**d**) the estimated positions of the Yellow Warbler over the entire 11 min experiment, relative to ARUs A, B, C and D.

We conducted the experiments in June and early July of 2017 at Pennsylvania State Game Lands 249, Heidlersburg, Adams County, Pennsylvania (39°56′14.6″ N, 77°10′38.6″ W). All the experiments were conducted between 6:00 and 9:00 a.m. The 140 hectare study site is mostly composed of old fields with mixtures of cold-season and warm-season grasses, hedges, scrub, and small wetlands. Most of the songbirds in the area sing from low shrubs and hedgerows, typically less than 5 m above the ground. The experiment included 30 replicates at locations on a 200 m grid to ensure a representative sample of the habitat and to reduce the likelihood of resampling birds between replicates. At each location, we deployed four SM3 ARUs (Wildlife Acoustics) on tripods 1.5 m off the ground in a 50 m × 50 m array centered on predetermined grid points. The recorders were fitted with Garmin GPS time-synch receivers to synchronize their clocks. Once the recorders were set up, we retreated to at least 150 m from the center of the array and waited at least 14 min before the drone flights to allow for a resettlement period to allow normal bird activity to resume following possible human disturbance. After the resettlement period, we deployed the drone from a base station at least 150 m away. We flew the drone in a direct line at 48 m above ground level (a.g.l.) and programmed it to autonomously hover 48 m a.g.l. over the center of the array for three minutes, to simulate an aerial point count as described in Wilson et al. [13]. After returning the drone to the base station, we continued recording bird song in the array from the ARUs for an additional four minutes. All the flights were preprogrammed using the Litchi App (VC Technology Ltd., London, UK) to ensure high

degrees of spatial precision. Each location was surveyed once, and adjacent locations were never overflown prior to when the experiment at that array took place, to avoid the potential for birds to become habituated to the drone.

### 2.2. Bioacoustic Analysis and Sound Localization

We used Audacity© version 2.3 to clip the recordings from each of the four ARUs to the exact same 11 min timespan (to nearest millisecond), which included 4 min before the drone flight, 3 min while the drone was overhead, and 4 min after the drone flight. We used 4 min before and after the drone flight so that we could exclude the minute immediately before and after the drone flight, during which the drone was audible on the recordings as it approached and departed the array (Figure 2). This allowed us to isolate 3 × 3 min blocks for "before" (minutes 1–3), "during" (minutes 5–7) and "after" (minutes 9–11) the drone flight. The drone flights from the base station to the array were always under one minute; hence, the drone was never airborne in minutes 1–3 (before), or 9–11 (after).

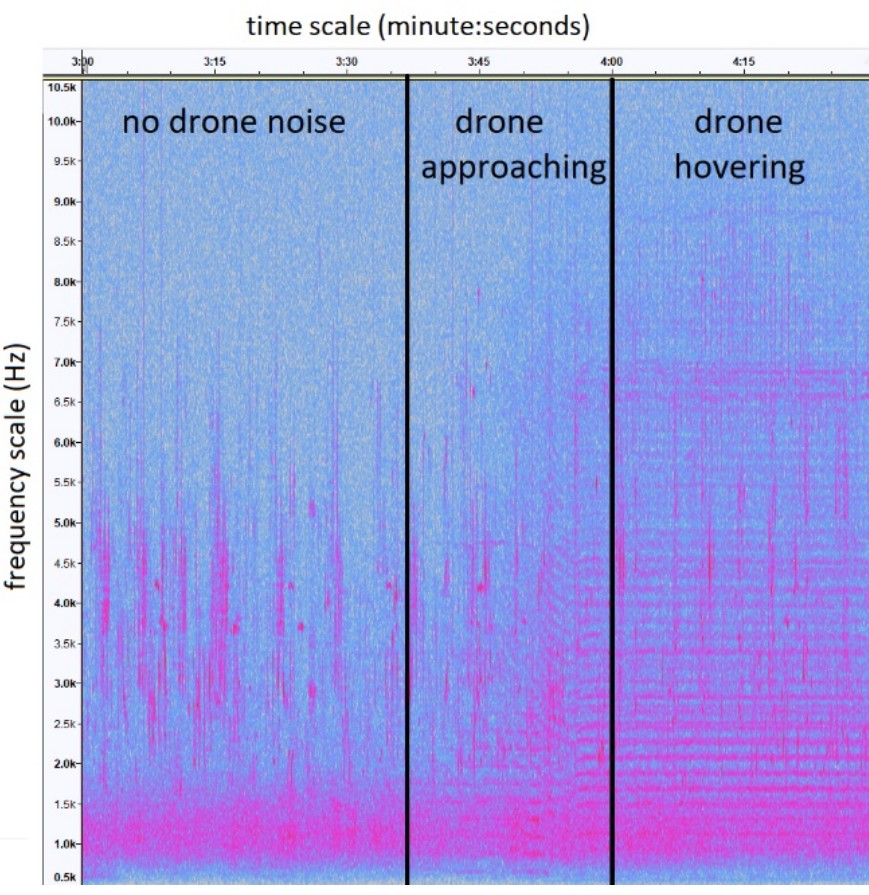

**Figure 2.** Spectrogram of a recording around the time the drone approached and hovered over the array (minutes 3 to 4.5), showing the potential effects of the drone noise on masking bird songs. Bird songs are the vertical lines in the 2.5 to 7 kHz frequency range, drone noise is the horizontal lines evident when the drone was approaching (minutes 3:37 to 4) and hovering overhead (after minute 4).

We focused our data analysis on seven songbird species that were known to be present on the site in moderate to high densities: American Robin *Turdus migratorius*, Yellow Warbler *Setophaga petechia*, Common Yellowthroat *Geothlypis trichas*, Field Sparrow *Spizella pusilla*, Song Sparrow *Melospiza melodia*, Indigo Bunting *Passerina cyanea*, and Northern Cardinal *Cardinalis cardinalis*. All seven species are discontinuous singers [19], that is, each song bout by each species is separated by silent phrases. Some other abundant species were not included in the study because we could not reliably identify individual birds in the audio recordings due to incessant continuous song types (e.g., Gray Catbird *Dumetella carolinensis*)

and localized high densities (Red-Winged Blackbird *agelaius phoeniceus*). For our seven study species, we identified all song bouts that were clearly visible in spectrograms from all four ARUs (Figure 1c). To identify the song bouts from each individual bird, we assigned a unique identification code to identify each individual bird based on location, song volume (visually assessed in spectrogram), song bout spacing, and unique characteristics of song.

The arrival time differences were measured manually from spectrograms in the program Raven Pro 1.5 (Figure 1c). The spectrogram settings were Hanning window with 512 samples and 89% overlap. We manually measured time differences between the four recordings because we found that using a sound correlator [18] led to some spurious results, which we attribute to the temporally overlapping songs of multiple species or individuals, and potentially to erroneous correlation between drone noise detected at the four ARUs. The time differences and air temperature at the time of the experiment (logged by the ARUs) were then input into the Sound Finder spreadsheet to localize each bird, with the locations given in UTM coordinates. Each individual bird was localized for every song bout throughout the 11 min experiment, so that the bird's movement and song output could be tracked through space and time.

While listening to the recordings to locate song bouts from our target species, we observed an apparent increase in Northern Cardinal "tik" contact/alarm calls when the drone was overhead. Because these calls are quieter than songs, and therefore difficult to localize because many are not detected on all four ARUs, we used a different approach to determine whether the rates of these calls changed in the presence of the drone. We used Kaleidoscope software [20] to automatically detect Northern Cardinal "tik" calls in recordings from a single ARU (A) from each array. To train Kaleidoscope, we developed a training set of 10 recordings of "tik" calls. All identified calls were verified aurally, to exclude calls from other species.

*2.3. Statistical Analysis*

The statistical analysis was restricted to the detection of birds within 50 m of the center of the array. This includes the area closest to the hovering drone, and hence includes the bird most likely to be disturbed by the sight or sound of the drone. Further, birds more distant than 50 m from the array center were less likely to be detected on all four ARUs, especially when the drone was overhead, potentially masking vocalizations from more distant birds (Figure 2). To test whether the birdsong detections varied by time, we constructed linear mixed-effects models (GLMMs) using the 'glmer' function in R 3.6.2© package lme4 [21]. Models were constructed separately for each species. We conducted two sets of analyses: (1) the probability of detecting an individual bird in each minute; and (2) the singing rate in each minute. For the first analysis, the response variable was whether an individual bird was detected during that minute, modeled with a binomial error distribution. We constructed eight models: a null model with the individual bird as a random effect (model 1), a model with distance to bird from the center of the array as a covariate (model 2); three models with time effects (models 3–5); and three models with time effects plus distance (models 6–8). The three time effects were: individual minutes (models 3 and 6); the 3 min periods "before", "during" and "after" (models 4 and 7); and "before" and "after" but with effects varying by minute during the drone flight (models 5 and 8; Figure 3). Time periods and minutes were included as factors. All eight models included an individual bird as a random effect. The models were compared using AIC, the size of the effects was determined using odd-ratios, and the significance was determined using Wald tests.

**Figure 3.** Chart showing models used to test whether bird song output is related to the drone's presence. All models include individual bird as a random effect. The null model and model 2 do not include time effects. Time effects of other models shown by numbered sequences. Note that minutes 4 (drone approaching) and 8 (drone departing) of the experiment were not included in the statistical analysis.

We used the same eight model structures illustrated in Figure 3 to test whether the singing rate (number of song bouts per minute) of each bird varied according to time-period, this time with a Poisson error distribution. The same Poisson model was used to test whether the Northern Cardinal's "tik" contact/alarm call showed time-dependent patterns consistent with a response to the drone flight.

## 3. Results

### 3.1. Probability of Singing Bird Detection

We identified 8303 song bouts across the seven study species, which we initially estimated to belong to 467 different individual birds (Table 1). However, these values are likely overestimates because ~50% of the song bouts were localized to distances over 100 m from the array centers, and hence duplication of individuals between adjacent arrays was likely. To provide a more accurate analysis, we excluded song bouts that were localized to distances greater than 100 m from the array centers. Further, many song bout detections were either not localizable (typically because they were not clear enough in all 4 spectrograms), or feature large localization errors (greater than 5 m) and were excluded from data analysis.

**Table 1.** Total number of song bouts detected, number of bouts localized within 100 m and 50 m of the array centers, and number of individual birds within the 50 m radius.

| | Total Song Bout Detected | Localized Song Bouts | | Birds within 50 m of Array Centers |
| --- | --- | --- | --- | --- |
| | | Within 100 m of Array Centers | Within 50 m of Array Centers | |
| American Robin | 1597 | 858 | 331 | 18 |
| Yellow Warbler | 1643 | 759 | 529 | 42 |
| Common Yellowthroat | 734 | 382 | 137 | 13 |
| Field Sparrow | 1388 | 532 | 388 | 31 |
| Song Sparrow | 1661 | 963 | 470 | 42 |
| Indigo Bunting | 611 | 357 | 241 | 14 |
| Northern Cardinal | 669 | 294 | 189 | 24 |
| All Seven Species | 8303 | 4145 | 2285 | 184 |

We found that the number of singing birds detected 0 m to 50 m from the array center appeared less variable over time than the number detected 50 m to 100 m from the center (Figure 4). The song bout detections showed a decrease in the three minutes when the drone was overhead, but most of this decline was from more distant birds, while the number of detections within 50 m was less variable over time (Figure 4). However, there were clear differences among the seven study species. For three species, Common Yellowthroat, Song Sparrow, and Indigo Bunting, the patterns did not suggest a significant response to the drone. Northern Cardinal detections increased during the drone flight, and the increased song output continued 4 min after the drone had departed (Figure 4). There were overall reductions in the number of singing American Robins, Yellow Warblers, and Field Sparrows when the drone was overhead, but most of this apparent decrease was from birds more than 50 m from the array center (Figure 4), hence further away from both the drone and ARUs.

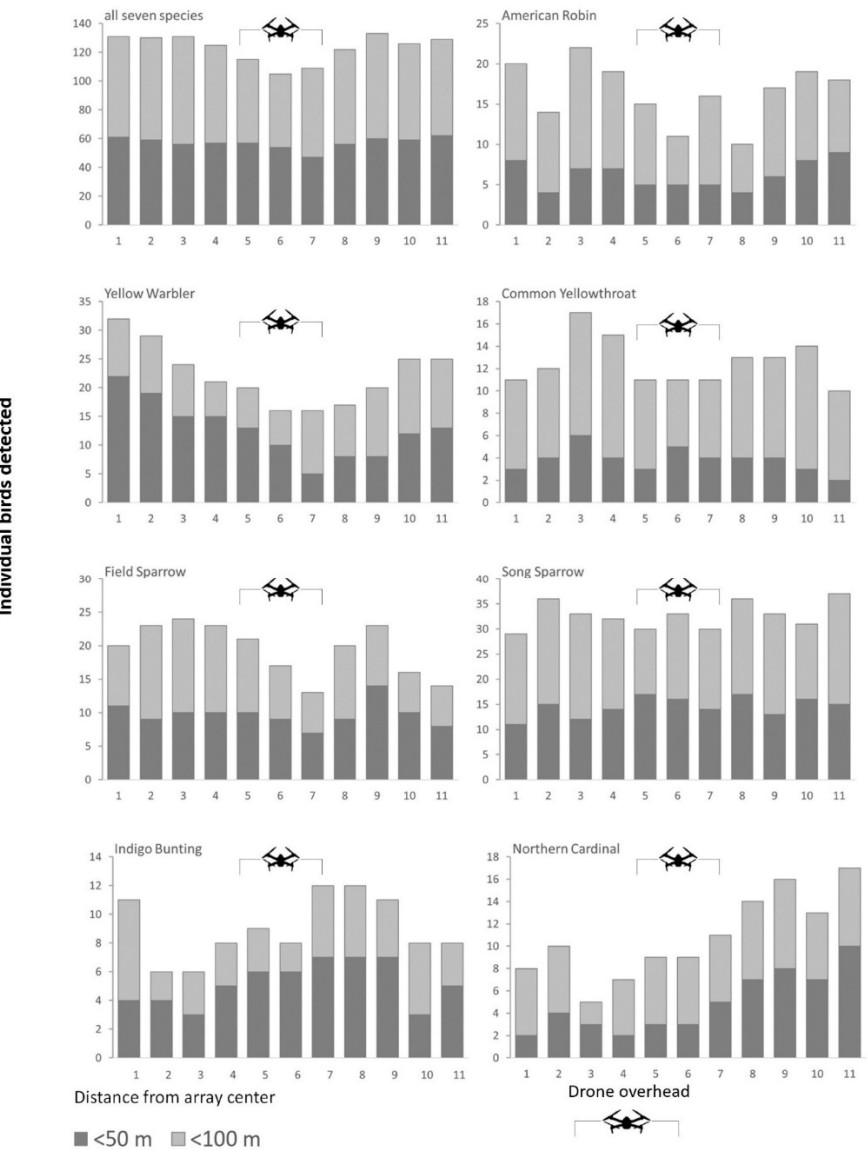

**Figure 4.** Stacked bar charts of the number of individual singing birds detected by minutes within 50 m and 100 m of the array center.

For birds within 50 m of the array center, the null model (no time effects) GLMM demonstrated the best fit (lowest AIC) for five of the seven species, indicating no evidence of an effect of the drone on song detections within a 50 m radius (Table 2). The null model

was also the best model when the data for all seven species were combined. The two exceptions were the Yellow Warbler and the Northern Cardinal. For the Yellow Warbler, model 5 was the best fit (Table 2), which indicates that song output decreased when the drone was overhead and that the effect increased over time, such that there was a non-significant (Wald-test; $p = 0.311$; Table 3) 34% decrease in detections in the first minute, a significant 54% ($p = 0.035$; Table 3) decrease in the second minute, and a highly significant 81% ($p = 0.001$; Table 3) decrease in the third minute, when compared with the period before the drone flight (Table 3). The song detections remained 51% lower after the drone had departed ($p = 0.013$; Table 3). For the Northern Cardinal, the drone effect was also significant, with model 4 (differences in detections between the three periods) the best model (Table 2). However, the results for Northern Cardinal were the opposite of those for the Yellow Warbler—cardinal song detections increased by a non-significant 46% ($p = 0.45$) in the 3 min period when the drone was overhead and increased almost five-fold ($p < 0.001$) after the drone had departed (Table 3).

**Table 2.** AIC values for the eight models for each species, for probability of detection. ΔAIC is compared with the null model (model 1). Gray shading indicates the lowest AIC (or joint lowest if ΔAIC less than 2).

| | **Model (See Figure 3)** | | | | | | | |
|---|---|---|---|---|---|---|---|---|
| **Species** | **1 (Null)** | **2** | **3** | **4** | **5** | **6** | **7** | **8** |
| | **AIC** | | | | **ΔAIC** | | | |
| American Robin | 194.8 | 1.8 | 11.1 | 2.6 | 6.4 | 13.0 | 4.5 | 8.3 |
| Yellow Warbler | 451.5 | 1.9 | −5.3 | −6.8 | −8.0 | −3.4 | −4.9 | −6.1 |
| Common Yellowthroat | 131.8 | 0.5 | 11.8 | 3.0 | 6.7 | 12.8 | 4.0 | 7.7 |
| Field Sparrow | 316.5 | 1.9 | 6.6 | 3.2 | 6.2 | 8.5 | 5.2 | 8.1 |
| Song Sparrow | 400.6 | 0.2 | 14.0 | 3.0 | 6.9 | 14.1 | 3.2 | 7.1 |
| Indigo Bunting | 157.6 | 1.4 | 7.9 | 2.2 | 5.4 | 9.3 | 3.6 | 6.8 |
| Northern Cardinal | 224.1 | 2.0 | −2.7 | −11.7 | −8.6 | −0.7 | −9.8 | −6.7 |
| All Seven Species | 1898.6 | 0.7 | 11.4 | 1.3 | 4.1 | 12.2 | 2.0 | 4.8 |

**Table 3.** Results of GLMMs for models showing significant effects of the drone on detections of birds within 50 m of the array centers. Estimates (Est) are odd-ratios relative the first time period (either minute 1, or period 1). LL and UL are lower and upper confidence limits, and *p*-values are from Wald tests.

| | | **Yellow Warbler** | | | |
|---|---|---|---|---|---|
| | | | **Est** | **LL** | **UL** | *p* |
| | Intercept | | 0.665 | 0.407 | 1.087 | 0.103 |
| | | Minute 5 | 0.662 | 0.297 | 1.472 | 0.311 |
| During | { | Minute 6 | 0.438 | 0.189 | 1.017 | 0.054 |
| | | Minute 7 | 0.189 | 0.066 | 0.495 | 0.001 |
| After | | Minutes 9–11 | 0.483 | 0.271 | 0.860 | 0.013 |
| | | **Northern Cardinal** | | | |
| | | | **Est** | **LL** | **UL** | *p* |
| | Intercept | | 0.111 | 0.050 | 0.248 | <0.001 |
| During | | Minutes 5–6 | 1.462 | 0.544 | 3.925 | 0.450 |
| After | | Minutes 9–11 | 4.888 | 1.977 | 12.084 | <0.001 |

### 3.2. Singing Rate

The null model (no time effects) was the best model for the singing rate of four species: Yellow Warbler, Field Sparrow, Song Sparrow, and Common Yellowthroat (Table 4). For the American Robin, model 7 (distance and 3 min period time effects) was the best fit, but none of the estimates for time periods were significant (Table 5). For the Indigo Bunting,

model 4 was the best fit, but the AIC was only marginally lower than that of the null model, and none of the estimates for time periods were significant (Table 5). We conclude that there was no evidence that the singing rate changed in response to the drone for six of the seven species. Once again, the results for the Northern Cardinal were anomalous, with a significant drop in singing rate in minute 6 ($p = 0.019$; Table 5), although the time-effect model (5) did not appreciably improve the model's fit over the null model (Table 4).

**Table 4.** AIC values for the eight models for each species, for singing-rate. ΔAIC is compared with the null model (model 1). Gray shading indicates the lowest AIC (or joint lowest if ΔAIC less than 2).

| | **Model (See Figure 3)** | | | | | | | |
|---|---|---|---|---|---|---|---|---|
| **Species** | **1 (null)** | **2** | **3** | **4** | **5** | **6** | **7** | **8** |
| | **AIC** | | | | **ΔAIC** | | | |
| American Robin | 264.1 | −4.0 | 7.2 | −0.4 | 1.8 | 2.5 | −5.5 | −4.0 |
| Yellow Warbler | 492.2 | 2.5 | 10.8 | 2.7 | 6.5 | 12.7 | 4.6 | 8.5 |
| Common Yellowthroat | 146.4 | 1.6 | 3.6 | 3.8 | 6.0 | 5.6 | 5.3 | 7.7 |
| Field Sparrow | 383.3 | 1.6 | 15.1 | 3.6 | 7.3 | 16.7 | 5.2 | 9.0 |
| Song Sparrow | 468.1 | 2.0 | 13.7 | 3.3 | 6.7 | 15.7 | 5.2 | 8.6 |
| Indigo Bunting | 202.5 | 1.7 | 10.3 | −0.1 | 3.4 | 12.2 | 1.8 | 5.4 |
| Northern Cardinal | 193.0 | 1.7 | 6.6 | 2.3 | −0.3 | 8.5 | 4.1 | 1.6 |
| All Seven Species | 2152.2 | −1.8 | 7.7 | 1.1 | 4.0 | 6.0 | −0.4 | 2.5 |

**Table 5.** Results of GLMMs for models showing significant effects of the drone on singing rates of birds within 50 m of the array centers. Estimates (Est) are odd-ratios relative the first time period (either minute 1, or period 1). UL and UL are lower and upper confidence limits, and *p*-values are from Wald tests.

| **American Robin** | | | | |
|---|---|---|---|---|
| | | Est | LL | UL | P |
| Intercept | | 6.124 | 4.274 | 8.774 | <0.001 |
| During | Minutes 5–7 | 0.847 | 0.605 | 1.188 | 0.337 |
| After | Minutes 9–11 | 1.240 | 0.922 | 1.669 | 0.155 |
| Distance | | 0.984 | 0.972 | 0.995 | 0.005 |

| **Indigo Bunting** | | | | |
|---|---|---|---|---|
| | | Est | LL | Est | LL |
| Intercept | | 4.846 | 3.786 | 6.204 | <0.001 |
| During | Minutes 5–7 | 0.711 | 0.501 | 1.009 | 0.056 |
| After | Minutes 9–11 | 0.728 | 0.511 | 1.037 | 0.079 |

| **Northern Cardinal** | | | | |
|---|---|---|---|---|
| | | Est | LL | Est | LL |
| Intercept | | 3.792 | 2.494 | 5.764 | <0.001 |
| | Minute 5 | 0.759 | 0.351 | 1.640 | 0.483 |
| During | Minute 6 | 0.231 | 0.068 | 0.783 | 0.019 |
| | Minute 7 | 0.942 | 0.506 | 1.754 | 0.851 |
| After | Minutes 9–11 | 0.782 | 0.491 | 1.247 | 0.302 |

| **All Seven Species** | | | | |
|---|---|---|---|---|
| | | Est | LL | Est | LL |
| Intercept | | 3.736 | 3.200 | 4.363 | <0.001 |
| During | Minutes 5–7 | 0.907 | 0.804 | 1.024 | 0.115 |
| After | Minutes 9–11 | 0.968 | 0.862 | 1.087 | 0.586 |
| Distance | | 0.996 | 0.991 | 1.000 | 0.055 |

*3.3. Cardinal Calls*

A total of 1148 Northern Cardinal "tik" calls were detected on recordings from ARU "A", from 23 of the 30 arrays. The calling rate increased from an average of 3.21/min (se 1.39) during minutes 1 to 4 to 5.96/minute (se 1.11) in minutes 4 to 9, then declined to 2.60/min (se 0.79) in the last two minutes of the experiment (Figure 5). The calling rates were significantly higher in minutes 5 to 9 compared with minute 1 (Wald tests, $p < 0.001$). It should be noted that there appeared to be an inflection point in the call rate at about 3 min 30 s, which is around the time that the drone was first audible in the recording as it approached the array (Figure 5).

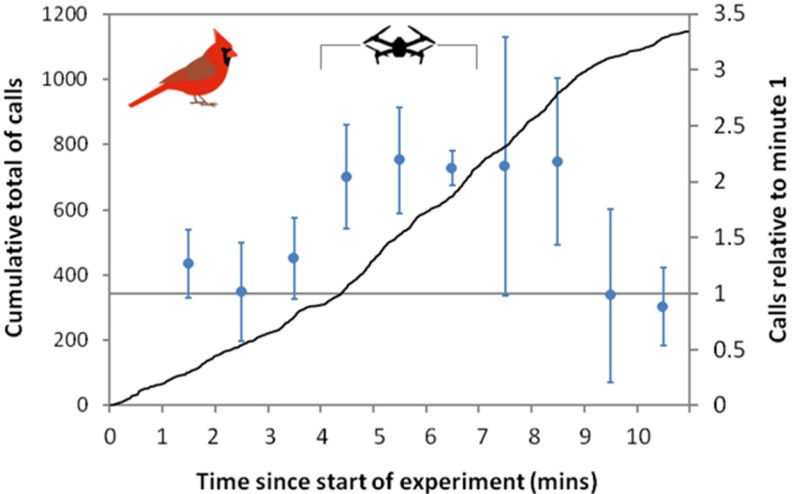

**Figure 5.** Calling rate of Northern Cardinals from a single ARU across all 30 arrays. Black line shows the cumulative total of calls and blue circles show GLMM coefficients for each 1 min interval, relative to minute 1, with 95% confidence limits.

## 4. Discussion

Drawing on multiple lines of evidence, we conclude that a drone hovering at 48 m above ground level for 3 min causes little overall change in songbird output in the area below the drone, within a 50 m radius. However, there were significant impacts of the drone on the detection of the songs of two species: the Yellow Warbler, which saw a substantial drop in song detection while the drone was overhead, and the Northern Cardinal, which saw a significant increase in song detection, especially after the drone's departure. The pronounced increase in Northern Cardinal contact/alarm calls during and following the drone flights provides further evidence that the vocal behavior of this species was affected by the drone's presence.

The fact that we found that responses to the drone were varied among a small group of passerines is intriguing. Differential responses to anthropogenic noise disturbance have been detected by others [22–24]. The differences have, in part, been attributed to whether the vocalization frequency of a bird species overlaps with the frequency of the noise disturbance [23], presumably because birds would not waste energy vocalizing if it was likely to be masked by other noise. However, of our seven study species, two emitted relatively low-frequency songs (American Robin and Northern Cardinal; peak sound pressure 2 to 4 kHz), the Field Sparrow songs were slightly higher (peak sound pressure 3 to 5 kHz), and the other birds studied had higher-frequency songs (peak sound pressure 4 to 8 kHz) [25], so we found no discernible association between sound frequency and effects of the drone. Being a small vehicle, the noise emitted by a DJI Mavic Pro is of higher frequencies than larger drones, and with a broad spectrum from less than 500 Hz to over 10,000 Hz [26], which overlaps with the songs of all the species included in our study. The increased singing and calling of Cardinals is indicative of their agitation at the

drone noise. The Northern Cardinal is a species that has been shown to increase its song frequency in response to anthropogenic noise [27,28].

It should be noted that we chose the DJI Mavic Pro for this study because of its suitability for conducting aerial bioacoustics surveys: it is small and therefore quieter than the larger vehicles that are typically used in wildlife research. The manufacturer reports sound pressure (SPLa) of 70 dB at 1 m [29], which would attenuate to around 36 dB at ground-level directly below a drone hovering at 48 m, and around 33 dB at ground level at the edge of the 50 m radius. In a review of anthropogenic disturbance affects, it was found that impacts on bird song, abundance, reproduction, and stress hormone levels were typically found when anthropogenic noise levels exceeded 45 dB SPL [30]. Therefore, while not wishing to downplay the noise pollution emitted from small drones, the noise levels that birds were subjected to in our experiment were modest in comparison to some other sources of anthropogenic noise. However, the relatively high frequency of drone noise compared to most forms of anthropogenic noise pollution, which is loudest below 2 kHz [31], results in more overlap with the frequency of most bird songs; hence, it could be that drone noise disturbance to birds is a function of its frequency, rather than loudness.

It has previously been shown that prolonged exposure to drones has an increased disturbance effect [16]. Our results support this, with impacts most evident after 2 min of exposure for both the Yellow Warbler and Northern Cardinal. If drones are to be used for bioacoustics surveys for birds, we recommend that very short-duration counts are used. Not only does this reduce disturbance, but it also allows multiple counts to be made on a single drone battery. If short-duration counts are considered inadequate, for example if the probability of detecting a bird is too low, an alternative approach would be to perform repeated counts over the duration of a day, or perhaps over several days. This would then provide a sampling framework that is amenable to estimating species presence or abundance using occupancy models [32,33] or spatially replicated N-mixture models [34]. An added advantage of using drones for aerial bioacoustics surveys under those circumstances is that they are highly replicable, both in terms of location and duration (due the ability to precisely program missions), and because audio recordings provide a permanent record of bird song that can be analyzed by multiple observers after the fact [35].

A limiting factor in our study is that despite the very large number of song bouts analyzed, the sample sizes for the number of individual birds for some of our study species were small, which may have reduced our ability to detect more modest responses. However, given the fact that many of the birds tracked were found to sing throughout the time that the drone was overhead, and that there was no evidence that singing rates declined for six of our seven species, we can be certain that the effects were very limited for five of the seven species. It should also be noted that most of the species in our study sing from relatively low vegetation (at least at our study site, pers. obs.); the Indigo Bunting being an exception. Therefore, we cannot generalize our results to species that sing from more elevated perches, which would be closer to the drone and perhaps more susceptible to disturbance. Clearly, further studies of the impact of drones on bird singing behavior are merited to determine whether our results hold for a wider range of species, and different habitat types.

Measurement error, especially with respect to the sound localization process, could also have impacted our results. Limiting the statistical analysis to within 50 m of the array center will have reduced localization errors because errors increase with distance from the array [18,36], but even so, we should expect that errors of 5 to 10 m were commonplace, even if the mean error was small [18]. Given that our study did not aim to estimate population densities—a common application of sound localization—the effects of these errors on our findings may be limited to whether or not individual birds were included in the statistical analysis (i.e., whether were they <50 m from array center). We do not think that the erroneous inclusion or exclusion of a few individuals would be sufficient to change our findings, especially as those marginal birds were furthest from the drone, and hence the least likely to be affected by its presence.

It is possible that our ability to individually identify birds based on their song pattern and location was not always accurate. Anecdotally, we observed that Yellow Warblers may have shown song-pattern plasticity [37], which could potentially have resulted in us over-estimating the number of different birds. However, because we used sound localization to track the birds in space and time, we believe that cases of confusion due to individual birds song-switching would be limited to a few individuals at the most.

Our study focused on assessing the effects of drone noise on bird song for a specific vehicle type and altitude, which was due to our interest in using drones for aerial bioacoustics. However, our results hold more general interest for those concerned about the disturbance effects of drones on wildlife, and to our knowledge, ours is the first study to focus specifically on how drones affect bird vocalizing behavior.

## 5. Conclusions

We found that a small quadcopter drone affected the singing behavior of two out of five songbirds in our experiments. This suggests that researchers who propose to use drones to record bird vocalizations should do so cautiously by using the quietest vehicle that would suit their needs, and by flying the drone at a height that will minimize the possibility of noise disturbance. The fact that we did find some bird behavioral responses to the drone in our study gives us reason for pause with respect to using drones for airborne bioacoustics. However, there has been a rapid improvement in technology to reduce unwanted drone noise, even in the time since our study. More recent DJI Mavic models are substantially quieter, with refined propeller shapes reportedly reducing sound pressure by 4 dB, which represents a reduction of about 60% in noise [29]. With the advent of quieter drones, we hope that concerns about noise pollution impact on wildlife will diminish, but we urge continued vigilance with regards to the behavioral impact of using drones to study wildlife.

**Author Contributions:** Conceptualization, A.M.W.; methodology, A.M.W., K.S.B., J.L.G., C.J.K. and M.F.W.; formal analysis, A.M.W., K.S.B. and M.F.W.; investigation, A.M.W., K.S.B., J.L.G., C.J.K. and M.F.W.; resources, A.M.W.; data curation, A.M.W., K.S.B., J.L.G., C.J.K. and M.F.W.; writing—original draft preparation, A.M.W.; writing—review and editing, A.M.W. and K.S.B.; visualization, A.M.W.; supervision, A.M.W.; project administration, A.M.W.; funding acquisition, A.M.W., K.S.B. and J.L.G. All authors have read and agreed to the published version of the manuscript.

**Funding:** This research was funded by the Gettysburg College Cross-Disciplinary Science Institute (X-SIG).

**Institutional Review Board Statement:** Our experiments were approved by the Gettysburg College Institutional Animal Care and Use Committee (IACUC), IACUC #2017S4.

**Informed Consent Statement:** Not applicable.

**Data Availability Statement:** Data are available from A.M.W., on request.

**Acknowledgments:** We thank the Pennsylvania Game Commission for permitting us to use State Game Lands 249 for the experiments. State Game Lands 249 are on the lands of the Susquehannock people. Gettysburg College students Taylor Derick, Arthur Garst, and Samantha Trueman helped with data analysis.

**Ethics Statement:** The drone was operated by the US Federal Aviation Authority (FAA) Part 107 license holder A.M.W., and in accordance with FAA guidelines.

**Conflicts of Interest:** The authors declare no conflict of interest.

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
