# Peer review of "Species-Specific Responses of Bird Song Output in the Presence of Drones"

_drones, doi:10.3390/drones6010001_

Round 1

Reviewer 1 Report

This paper addresses an interesting issue on the effect of drone flight on bioacoustics research, in particular, the main purpose is how noise from drones affects bird vocalization behavior. There are still few examples of such studies, and the content of this paper can be evaluated as meaningful. Although I believe the paper will be of interest to the readership, I would suggest a few revision before ready for publication.

<Major comments>

  1. Please add a description of how high the observed birds were. The impact depends on whether most were near the ground or, in some cases, near the drone.

  1. As with comment 1, please specify whether the flock of birds has moved during the observation.

  1. Does "<100 m" in Figure 3 mean from 50 m to 100 m ?

  1. It seems that the descriptions on lines 228-229 and the data shown in Figure 3 are inconsistent. Perhaps the legend in Figure 3 is reversed (light gray is "<100 m", dark gray is "<50 m"?).

  1. On line 247, if you mention masking, it is better to show the noise level of the drone described here. In addition, please indicate the average sound pressure level of the observed birds' voice.

  1. This paper ends with a discussion section. It would be easier for readers to understand the authors' claims if a conclusion section is provided.

<Minor comments>

  1. Is Audacity on line 121 the audio software (https://www.audacityteam.org/)? The copyright and version should be specified.

  1. For R on line 177, the copyright and version should be specified.

  1. In the captions of Tables 3 and 5, "UL and UL" -> "LL and UL"

  1. On line 285, Table 3 -> Table 5

  1. For "broad spectrum" on line 329, please specify the frequency range.

Reviewer 2 Report

The authors describe a well-designed study of effects of UAV noise on small birds.  This would be a valuable contribution to the literature.  The manuscript would would benefit from adding a few examples of how the observed behavioral changes might influence bird ecology, either in the introduction or discussion.   The methods are thoughtful and appear sound.  The statistical methods are appropriate and clearly described.  The figures and tables are well-designed.  The conclusions are supported.  My main comments are to clarify some aspects of the methods and the results.  Please find specific comments below.

Abstract

l 13. Suggest changing “50 meters above ground level” to either “about 50 meters above ground level” or “48 meters above ground level” as the paper later consistently describes the altitude of the drone to be 48 meters above ground level.

l 34–35. Suggest changing “surveys large birds” to “surveys of large birds”.

Materials and Methods

At what time of day were the flights made?  How could this influence the results?  

How many times did the researchers revisit a given location throughout the season?

What is the amount of time between each treatment period, especially when array sites were adjacent or even possibly overlapping?  While it is possible that future drone bioacoustics studies will deal with some degree of habituation in the birds, I would be careful to not complete treatment periods in close spatial and temporal proximities. I could imagine a potential situation where birds would already be singing or calling at a different frequency because of the last treatment, while the authors completed the new treatment in an area that could pick up a bird from the last treatment. This could potentially confound the data as there would be less change in a bird’s behaviors with the introduction of the drone if they are still reacting to the last treatment. A greater confidence in identifying an individual bird’s song could also allow the authors to throw out any individual in a subsequent treatment that may have been present in a previous treatment and may still be responding to that previous treatment.

Along with defining the reset period between treatments, I would also ask the authors to provide further explicit definition of when the control period (first four minutes of each experimental period) took place. While describing the ten minutes to allow the birds to settle after placing the ARUs in the field, it is unclear if the control period was part of the ten minutes allotted to resettlement, or if the initial four minutes of the experimental period began after the ten-minute resettlement period. 

l 89. Unclear what 74% refers to.  Is it 74% of points?

l 104. What is the total area of the study site?  This information would be helpful because the authors later describe efforts to minimize double-sampling of individuals.  If the study site is only 200m across, then it seems very likely that some individuals would be sampled more than once. 

l 111. “After the resettlement period we deployed the drone” sounds like the first four control minutes were included as part of the ten minutes for resettlement. Further clarification of this time delineation would be helpful.

Paragraph starting on line 121. Please check if dash is required between number and minutes (e.g. “4-minutes”)

Paragraph starting on line 132. Please check if scientific names should be italicized.

Figure 2. This is an excellent figure that clearly shows the model structures.  Names for models 3 and 7 start with periods—please remove or explain the meaning.

l 187. Please specify if time variables were specified as numeric or factor variables.

Assuming there are raptor species in the study area, I am also interested if the authors witnessed an increase in abundance of raptors during the control periods, which may use the ARUs as perches. Based on the description of the study design however, I would assume the ARUs were not deployed long enough to become potential raptor perches during treatments. 

Results

The main results of section 3.1 appear to be those in the paragraph starting on line 255, so I would recommend leading the results section with that paragraph.  It is uncurl if the results paragraphs prior to that add anything.

l 212. Use of ms could lead a reader to think of milliseconds rather than meters.  Is m more appropriate?

l 222. Suggest changing “less variable time” to “less variable over time”.

Table 1. This table is hard to interpret without the information requested for line 104.

Figure 3.  Please changing signing to singing in title.  The lower left panel is missing the x-axis values.  The figure needs an x-axis title on at least one panel (or at the bottom of the figure to match formatting for the y-axis title).  Suggest moving legend (colors and drone image) down to provide room for x-axis title.

l 218-219.  It is unclear how the authors concluded that the number of singing birds from 0–50m were less variable than those at 50–100m.  To make this statement, they need to provide a quantitative assessment, possibly using statistics.

Regarding paragraphs starting on lines 218, 237, and 246.  More details are needed to help the reader understand the support for these statements.  In particular, 1) please state which results are based on GLMM models and which are based on some other quantitative assessment (for example, according to the methods, GLMMs were not used to study birds over 50m away), and 2) please refer to the model ranks to support your statements.  The authors describe the linear models in the paragraph starting on 255, but this information should be described when making statements in the prior paragraphs to help the reader assess the support for those statements.

l 220-226 and l 237-245 are identical. I suggest one of these groups of lines should be deleted. I would suggest deleting l 237-245 as they are missing the last part of the paragraph attached to l 220-226. 

l 246–251. The authors made the point in the Methods that the models would be restricted to birds within 50m, so it is no necessary to repeat it here.  Some of the rationale for limiting the data set to within 50m could be moved to the methods.

l 275. In the Table 3 description, suggest changing “UL and UL” to “LL and UL”.

l 293. In the Table 5 description, suggest changing “UL and UL” to “LL and UL”.

l 277–287. This paragraph gives a clear description of the results.

l 297. Change increase to increased.

l 302. Change to minutes (plural)

Discussion

l 331. Change to “shown”.

l 356. Suggest changing “that was amenable” to “that is amenable”.

l 367. Insert “the” before drone.

l 381. Change to “population estimates”

l 398. Change to “The fact”
